## Perspective

health and disease and epidemiology

COVID-19, population testing, modelling

**Author for correspondence:**
Julian Peto
e-mail: julian.peto@lshtm.ac.uk

# Weekly population testing could stop this pandemic and prevent the next

## Julian Peto

London School of Hygiene and Tropical Medicine, London, UK

JP, 0000-0002-1685-8912

The rapid spread of the SARS-COV-2 delta variant in the UK despite high vaccination coverage will inevitably accelerate when social restrictions end unless testing and contact tracing become much more effective. To minimize further social and economic damage, the effect on R of introducing weekly population testing as social restrictions are relaxed should be evaluated. The large increase in testing capacity required can be achieved with self-taken saliva samples analysed by RT-LAMP in local testing facilities. The costs and effectiveness can be evaluated in whole-city demonstration studies. A local population register in each city or district is essential to issue weekly invitations, manage sample collection, monitor results and achieve rapid notification of households and other contacts when a test is positive. In the UK, weekly test invitations should be managed, like vaccination invitations, by the NHS, with social and financial support for quarantined households to make self-isolation acceptable. A framework for effective population testing that had been established and evaluated during this pandemic could be rapidly reinstated to suppress the next pandemic while vaccines for a new and perhaps more deadly virus are developed and rolled out.

A leading modelling group warned in March 2021 that social restrictions might have to continue in the UK until vaccination roll-out is completed and perhaps beyond to control emerging SARS-CoV-2 variants, and concluded that new approaches are urgently needed to avoid further social restrictions [1]. Since then, the delta (B.1.617.2) variant has become the dominant strain in the UK and many other countries, and other increasingly infectious variants are emerging worldwide [2]. The rapid spread of the delta variant in the UK despite high vaccination coverage will accelerate when social restrictions end unless testing and contact tracing become much more effective. SARS-COV-2 vaccines were developed and tested with unprecedented speed, but there has been no equivalent effort to

establish and evaluate regular testing of the whole population except in Slovakia, where increasing prevalence was transiently reversed by two rounds of weekly national testing with curfew for anyone without a negative test [3]. Prevalence fell by 58% within a week, and a simulation calibrated to the Slovakian data confirmed the crucial role of quarantine for household members of those testing positive [3]. These observations support the suggestion that a month or two of weekly testing with high coverage and household quarantine might be an effective and much less economically damaging alternative to maintaining social restrictions to control the sharp recent increase in prevalence due to the delta variant as vaccination is rolled out [4]. Moreover, if SARS-CoV-2 can be controlled by weekly testing, the next pandemic, which may be caused by a much more deadly new virus, could probably also be suppressed while a vaccine is being developed and distributed. Establishing the local framework for population testing in a few cities and evaluating its impact on R during this pandemic should be prioritized for that reason alone. The system could be rapidly reinstated, together with testing for international passengers, at an early stage of a future pandemic before more infectious variants have evolved.

The British government accepted the case for regular population testing and announced Operation Moonshot in September 2020 but walk-in test centres achieved inadequate coverage, and when mass vaccination began the original aim of testing the whole population weekly was abandoned. Contact tracing by NHS Test and Trace (a misleadingly named organization staffed mainly by contractors and consultants with a budget of £37 billion) was contributing an estimated reduction in R of only 2–5% in October 2020 [5], and according to the UK Public Accounts Committee, there is still no clear evidence of its overall effectiveness [6]. In striking contrast, vaccination is being rolled out and evaluated with exemplary efficiency by the NHS by issuing personal invitations linked through local NHS patient records to attendance and subsequent hospital admissions and deaths. Demonstration studies in a few cities with financial and social support for quarantined households are needed for proper evaluation of the potential impact of mass testing. Integrated management of this experiment requires a local population register of names and contact details to issue weekly invitations, record test results and ensure that the majority of households and other contacts are notified rapidly when a case is detected [4]. In the UK these studies should be conducted by local NHS and public health authorities, comparing hospital admissions and deaths in the diminishing cohort who choose not to be tested and the regularly tested majority stratified by vaccination status. Few infectious cases would be missed by sensitive weekly testing so transmission chains within the regularly tested cohort would probably be stopped. Whether prevalence in the tested cohort would fall despite transmissions from the continuing epidemic in the untested minority and from international travellers can be established only by observing the effect as vaccination is rolled out and social distancing measures are relaxed. Requiring evidence of a recent negative test in some occupations and for access to public venues would both encourage compliance with testing and protect the regularly tested majority from those who choose not to be tested.

Weekly testing with good uptake and compliance with quarantine might increase the effectiveness of testing and tracing in the UK by an order of magnitude. Daily testing capacity is still only about 800 000 [6], less than 10% of the 10 million tests per day needed for weekly testing of the whole UK population. That increase in capacity could be achieved rapidly and economically by setting up local facilities in each city for RT-LAMP testing [4]. RT-LAMP on self-taken saliva samples can be almost as sensitive and specific as PCR on nasal or throat swabs [7,8] and does not require expensive equipment or highly trained staff. If sample tubes are bar-coded and delivered rapidly to a local test centre, results could be entered on NHS patient records within a few hours of sample collection. Lateral flow tests are useful for on-the-spot testing to identify the most infectious people but have several disadvantages for systematic population testing. Only 40% of infections detectable by PCR were detected by self-administered lateral flow tests in the mass testing pilot study in Liverpool [6], nasal swabs are more difficult to take than saliva samples, and incomplete self-reporting of results precludes rapid tracing of many contacts. Direct entry of all RT-LAMP test results to a local population register would ensure rapid access to households and other contacts when an infection is detected.

Sustained weekly testing in a whole city or district has never been tried despite support by experienced epidemiologists, statisticians, laboratory scientists and public health experts [4,9]. In the UK, the main obstacles to this experiment are the government's reluctance to transfer control and funding from NHS Test and Trace to local NHS and public health experts, the influential modellers who dismissed the idea of weekly population testing with effective household quarantine without having modelled it [10], and anti-testing campaigners who confuse the aims and ethics of population testing to stop an epidemic with population screening to detect and treat non-infectious diseases [11].

The UK response to this unprecedented public health crisis should include well-designed whole-city demonstration studies locally managed by the NHS to compare the costs and effects of weekly testing against the ineffective NHS Test and Trace national system managed by consultants and contractors. The results of this crucial experiment would certainly inform more realistic modelling, would probably prevent many deaths and further economic damage, and might prevent the next pandemic.

Data accessibility. This article has no additional data.

Competing interests. The author declares no conflict of interest.

Funding. No funding was received for this work.

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
