## [Peer Review File · Royal Society Open Science]

Review History

RSOS-210468.R0 (Original submission)

Review form: Reviewer 1

Is the manuscript scientifically sound in its present form?

Yes

Are the interpretations and conclusions justified by the results?

Yes

Is the language acceptable?

Yes

Do you have any ethical concerns with this paper?

No

Have you any concerns about statistical analyses in this paper?

No

Recommendation?

Accept with minor revision (please list in comments)

Comments to the Author(s)

This is an important contribution. Most western countries failed to combat the covid epidemic effectively due to dogma in a narrow and specialist scientific discipline combined with political inertia. In this review, Julian Peto suggests a trial / demonstration study that would help in convincing decision-makers and specialist scientists that a well designed continuous test-and-isolate strategy could eliminate infectious disease from a population in a real-world setting (from a western society with a relatively high prevalence of infection).

I support publication of this work after minor changes listed below:

* The test-and-isolate strategy should not be framed as an "alternative" to lockdown, which may be initially needed prior to effective test is developed

* It would be helpful if the differences between the proposed strategy and the current "targeted" approach to test-and-isolate would be explained more clearly. How much more testing is needed compared to the current approach in the UK?

* It would be helpful if the reason that the strategy is important in a future epidemic is more clearly stated. The temporal order at which different strategies become available in the best case is: indiscriminate lockdown (days) > test-(trace)-and-isolate (few weeks) > pharmaceuticals (months if already approved drugs work) > vaccines (~ 1 year), with vaccines not necessarily working, and in any case taking far too long to eliminate a fast-spreading disease before it causes a pandemic. This order is unlikely to change before the next pandemic, as technological challenges for making "instant" drugs or vaccines that are safe are enormous, and would require almost complete understanding of chemistry and biology.

The pandemic plan that was executed initially relied on vaccines and natural immunity, and accepted a large number of casualties as inevitable; it would be very important that a new plan would include approaches that work more rapidly.

Decision letter (RSOS-210468.R0)

Dear Dr Peto

On behalf of the Editors, we are pleased to inform you that your Manuscript RSOS-210468 "COVID-19: Stopping this pandemic and preparing for the next" has been accepted for publication in Royal Society Open Science subject to minor revision in accordance with the referees' reports. Please find the referees' comments along with any feedback from the Editors below my signature.

Please submit your revised manuscript and required files (see below) no later than 7 days from today's (ie 18-Jun-2021) date. Note: the ScholarOne system will 'lock' if submission of the revision is attempted 7 or more days after the deadline. If you do not think you will be able to meet this deadline please contact the editorial office immediately.

on behalf of Dr Julian Huppert (Associate Editor) and Nick Pearce (Subject Editor)
openscience@royalsociety.org

Associate Editor Comments to Author (Dr Julian Huppert):

Thank you for submitting this interesting paper. I'm sorry it has taken so long to arrange appropriate peer review.

This paper is suitable for publication with a few minor changes. Please respond to the referee's specific comments in the revised manuscript.

You may also want to consider commenting on efforts to do large-scale asymptomatic weekly testing, such as the University of Cambridge programme, which started last October.
<https://www.cam.ac.uk/coronavirus/stay-safe-cambridge-uni/asymptomatic-covid-19-screening-programme>

You may also wish to take the opportunity to update any relevant aspects of the paper given more recent events.

Reviewer comments to Author:
Reviewer: 1
Comments to the Author(s)

This is an important contribution. Most western countries failed to combat the covid epidemic effectively due to dogma in a narrow and specialist scientific discipline combined with political inertia. In this review, Julian Peto suggests a trial / demonstration study that would help in convincing decision-makers and specialist scientists that a well designed continuous test-and-isolate strategy could eliminate infectious disease from a population in a real-world setting (from a western society with a relatively high prevalence of infection).

I support publication of this work after minor changes listed below:

* The test-and-isolate strategy should not be framed as an "alternative" to lockdown, which may be initially needed prior to effective test is developed

* It would be helpful if the differences between the proposed strategy and the current "targeted" approach to test-and-isolate would be explained more clearly. How much more testing is needed compared to the current approach in the UK?

* It would be helpful if the reason that the strategy is important in a future epidemic is more clearly stated. The temporal order at which different strategies become available in the best case is: indiscriminate lockdown (days) > test-(trace)-and-isolate (few weeks) > pharmaceuticals (months if already approved drugs work) > vaccines (~ 1 year), with vaccines not necessarily working, and in any case taking far too long to eliminate a fast-spreading disease before it causes a pandemic. This order is unlikely to change before the next pandemic, as technological challenges for making "instant" drugs or vaccines that are safe are enormous, and would require almost complete understanding of chemistry and biology.

The pandemic plan that was executed initially relied on vaccines and natural immunity, and accepted a large number of casualties as inevitable; it would be very important that a new plan would include approaches that work more rapidly.

===PREPARING YOUR MANUSCRIPT===

===PREPARING YOUR REVISION IN SCHOLARONE===

Author's Response to Decision Letter for (RSOS-210468.R0)

See Appendix A.

Decision letter (RSOS-210468.R1)

Dear Dr Peto,

It is a pleasure to accept your manuscript entitled "Weekly population testing could stop this pandemic and prevent the next" in its current form for publication in Royal Society Open Science.

COVID-19 rapid publication process:

We are taking steps to expedite the publication of research relevant to the pandemic. If you wish, you can opt to have your paper published as soon as it is ready, rather than waiting for it to be published the scheduled Wednesday.

This means your paper will not be included in the weekly media round-up which the Society sends to journalists ahead of publication. However, it will still appear in the COVID-19 Publishing Collection which journalists will be directed to each week (<https://royalsocietypublishing.org/topic/special-collections/novel-coronavirus-outbreak>).

If you wish to have your paper considered for immediate publication, or to discuss further, please notify openscience_proofs@royalsociety.org and press@royalsociety.org when you respond to this email.

on behalf of Dr Julian Huppert (Associate Editor) and Nick Pearce (Subject Editor)
openscience@royalsociety.org

Appendix A

Dear Dr Huppert

Manuscript RSOS-210468

Responses to the referee's comments are given below. The manuscript has also been updated as you suggested to take account of developments over the 3 months since it was submitted, notably the emergence and rapid spread of the delta variant and relevant new references. A paragraph on RT-LAMP has also been added. The referee asked how many more tests would be needed, and the fact that the tenfold increase to almost 10 million tests per day can be achieved rapidly by establishing local RT-LAMP testing facilities is a crucial aspect of my case.

I hope you will agree that these amendments do not require further external review, as the new delta variant is now spreading exponentially. The aim of the piece is to provoke public discussion and hence rapid government action to conduct this experiment in a few cities, not to argue that mass testing would certainly be effective. Please let me know whether there is likely to be a long further delay before publication.

You suggested adding a section on the testing of staff and students at Cambridge. Many universities and various employers have introduced such schemes but the results do not reflect the coverage and impact that testing a whole city might achieve, as the populations tested are unrepresentative (young and educated) and are not protected from infection from the rest of the local community. A review of the results would have to be extensively referenced and would not materially affect the case for city-wide demonstration projects.

Best wishes

Julian Peto

Reply to reviewer comments to Author:

Reviewer: 1

Comments to the Author(s)

This is an important contribution. Most western countries failed to combat the covid epidemic effectively due to dogma in a narrow and specialist scientific discipline combined with political inertia. In this review, Julian Peto suggests a trial / demonstration study that would help in convincing decision-makers and specialist scientists that a well designed continuous test-and-isolate strategy could eliminate infectious disease from a population in a real-world setting (from a western society with a relatively high prevalence of infection).

I support publication of this work after minor changes listed below:

* The test-and-isolate strategy should not be framed as an "alternative" to lockdown, which may be initially needed prior to effective test is developed.

Author reply: "Lockdown" is ill-defined, so I have substituted "social restrictions" throughout. The amended text now makes it clear that I am suggesting that social restrictions could be stopped when most people have been vaccinated and population testing has been set up:

"These observations [the Slovakian data] support the suggestion that a month or two of weekly testing with high coverage and household quarantine might be an effective and much less

economically damaging alternative to maintaining social restrictions to control the sharp recent increase in prevalence due to the delta variant as vaccination is rolled out (4).”

* It would be helpful if the differences between the proposed strategy and the current "targeted" approach to test-and-isolate would be explained more clearly. How much more testing is needed compared to the current approach in the UK?

Author reply: I have inserted a new para on RT-LAMP which begins:

“Weekly testing with good uptake and compliance with quarantine might increase the effectiveness of testing and tracing in the UK by an order of magnitude. Daily testing capacity is still only about 800,000 (6), less than 10% of the 10 million tests per day needed for weekly testing of the whole UK population. That increase in capacity could be achieved rapidly and economically by setting up local facilities in each city for RT-LAMP testing (4).”

* It would be helpful if the reason that the strategy is important in a future epidemic is more clearly stated. The temporal order at which different strategies become available in the best case is: indiscriminate lockdown (days) > test-(trace)-and-isolate (few weeks) > pharmaceuticals (months if already approved drugs work) > vaccines (~ 1 year), with vaccines not necessarily working, and in any case taking far too long to eliminate a fast-spreading disease before it causes a pandemic. This order is unlikely to change before the next pandemic, as technological challenges for making "instant" drugs or vaccines that are safe are enormous, and would require almost complete understanding of chemistry and biology.

The pandemic plan that was executed initially relied on vaccines and natural immunity, and accepted a large number of casualties as inevitable; it would be very important that a new plan would include approaches that work more rapidly.

Author reply: If the system for population testing by RT-LAMP is established now it could be reinstated within weeks to control a future pandemic before lockdown becomes necessary when infection rates are still low and before new more infectious variants have evolved. A local RT-LAMP testing facility is very much easier to set up than the UK's very large regional PCR "lighthouse" labs. The relevant amended text is:

“Moreover, if SARS-CoV-2 can be controlled by weekly testing the next pandemic, which may be caused by a much more deadly new virus, could probably also be suppressed while a vaccine is being developed and distributed. Establishing the local framework for population testing in a few cities and evaluating its impact on R during this pandemic should be prioritised for that reason alone. The system could be rapidly reinstated, together with testing for international passengers, at an early stage of a future pandemic before more infectious variants have evolved.”